# Fetal Renal Duplicated Collecting System at 14–16 Weeks of Gestation

**DOI:** 10.3390/jcm12227124

**Published:** 2023-11-16

**Authors:** Nizar Khatib, Moshe Bronshtein, Gal Bachar, Ron Beloosesky, Yuval Ginsberg, Osnat Zmora, Zeev Weiner, Ayala Gover

**Affiliations:** 1Rappaport Faculty of Medicine, Technion-Israel Institute of Technology, Haifa 3200003, Israely_ginsberg@rambam.health.gov.il (Y.G.);; 2Department of Ob/Gyn, RAMBAM Health Care Campus-Israel, Haifa 3109601, Israel; gal.bachar13@gmail.com; 3Faculty of Social Welfare & Health Sciences, University of Haifa, Haifa 3200003, Israel; moshe.bronshtein@gmail.com; 4Sackler School of Medicine, Tel-Aviv-University, Tel Aviv 6997801, Israel; zmora.osnat@gmail.com; 5Neonatal Intensive Care Unit, Carmel Medical Center, Haifa 3200003, Israel; ayalagover@gmail.com

**Keywords:** renal duplication, ultrasound, early diagnosis

## Abstract

(1) Background: To examine the incidence of the prenatal diagnosis of the renal double-collecting system (rDCS) and describe its clinical outcome and associated genetic abnormalities. (2) Methods: This retrospective study included women who attended the obstetric clinic for early fetal anatomic sonography with findings of a renal DCS. Diagnosis was conducted by an expert sonographer using defined criteria. (3) Results: In total, 29,268 women underwent early ultrasound anatomical screening at 14–16 weeks, and 383 cases of rDCS were diagnosed (prevalence: 1:76). Associated abnormalities were diagnosed in eleven pregnancies; four had chromosomal aberrations. No chromosomal abnormalities were reported in isolated cases. Ectopic uretrocele and dysplastic kidney were diagnosed in 6 (1.5%) and 5 (1.3%) fetuses, respectively. One girl was diagnosed with vesicoureteral reflux and recurrent UTIs, and two boys were diagnosed with undescended testis. The recurrence rate of rDCS was 8% in subsequent pregnancies. (4) Conclusions: In light of its benign nature, we speculate that isolated rDCS may be considered a benign anatomic variant, but a repeat examination in the third trimester is recommended to assess hydronephrosis.

## 1. Introduction

A prenatal ultrasound (US) contributes to the increased detection of congenital renal anomalies. The duplication of the renal collecting system, known as the duplex kidney or double-collecting system (DCS), is the most common abnormality of the development of the renal tract detected prenatally [1,2], with an incidence of 1% in the general population [1]. It is characterized either by a renal unit consisting of two pelvicalyceal systems as a result of a premature division of a single ureteric bud or via two draining ureters due to two separate ureteral buds [3]. The duplication can be complete or partial, is reported more commonly in females (65% of cases), and occurs bilaterally in 20% of cases [4]. DCS is often asymptomatic and has no clinical significance. It is rarely associated with vesicoureteral reflux or recurrent urinary tract infections; however, urinary tract infections early in infancy or childhood have been described [4]. The anomaly is part of a large heterozygous group of renal malformations termed congenital anomalies of the kidney and urinary tract (CAKUT) [5]. The duplex collecting system can, at times, lead to medical complications, such as vesicoureteral reflux with associated periodic urinary tract infections, potentially causing kidney scarring and functional impairment. Other complications include multicystic dysplastic kidney and pelviureteric obstruction. However, the majority of urinary duplex anomalies are usually asymptomatic.

It may be associated with genetic conditions such as Turner syndrome [6] and Smith-Magenis syndrome and may be a part of the monogenetic diseases syndrome (e.g., branchio-otorenal syndrome and Fanconi anemia) [6].

The prenatal diagnosis of DCS has been previously reported during the late second and third trimesters [7,8].

Although renal DCS is prevalent and properly described in the medical literature, there is little documentation addressing its association with genetic syndromes. Searching in PubMED revealed two studies that addressed the association between isolated prenatally diagnosed duplex kidneys and genetic aberration [4,9]. This study was performed by cross-referencing the data of all women who performed amniocentesis with chromosomal microarray analysis, reported by the Israeli Ministry of Health, and the sonographic diagnosis of isolated DCS. The study’s conclusion was that isolated fetal DCS is not an indication for prenatal invasive tests. Among the limitations that were reported by the authors is the lack of a prenatal description of the sonographic appearance of the double-collecting system. Associations between isolated fetal renal DCS and chromosomal anomalies are still being determined. In this study, we report the incidence of the early prenatal diagnosis of renal DCS and describe its clinical outcome and associated genetic abnormalities.

## 2. Materials and Methods

This was a single-center retrospective chart review of all early prenatal anatomical screenings conducted by a single physician (M.B.) in an obstetrics clinic. Transvaginal sonographic examinations (Samsung USS-WS8El4U and Phillips BO3XP8) were performed at 14–16 gestational weeks in the lithotomy position after bladder emptying. The sonographer used his free hand to gently manipulate and move the fetus to achieve the requested images. The examination was performed with a UNI-U22 device (Philips Medical Systems, Bothell, WA, USA) with a 3–9 MHz vaginal transducer and Samsung WH80 3–10 vaginal probe. The exam included fetal biometry as well as a detailed sonographic scan aimed at detecting fetal structural malformations. Following the detection of the kidneys in the renal fossa, they were examined on both coronal and sagittal sections. In these sections, the kidney was normally visualized as an echogenic structure and easily distinguishable from the surrounding structures, with a hypoechogenic pelvis [Figure 1]. The diagnosis of a renal DCS was based on the following sonographic criteria: (A) visualizing the external surface shaped like a figure of eight, (B) the visualization of two renal pelvises like the figure of eight kidney sign (Figure 1) (Y-shaped pelvis was excluded via linear sagittal sections), and (C) the direct demonstration of two ureters (in some of the cases) (Figure 2 and Figure 3, Appendix A). Associated anomalies were noted and documented.

Follow up on fetuses diagnosed with DCS was acquired via direct communication with women whose fetuses had other abnormalities and from women returning for their next pregnancy in isolated cases. The follow up was based on parental reports and included specific questions regarding their child’s general health, development, hospital stays, urinary tract infections, postnatal ultrasounds, and urology follow up.

Ethical approval: this study was performed as per the ethical standards set by the Ethical Committee for Human Studies following the 1964 Declaration of Helsinki. This study was approved by the Ethics Committee for Human Study—Rambam Medical Center (approval number: 0138-20-RMB).

## 3. Results

There were 29,268 early anatomical scans performed between 14 and 16 gestational weeks in our obstetrics clinic. Cases were identified, and their data were extracted from the sonographic computerized database. Renal DCSs were diagnosed in 383 fetuses, with an incidence of 1.3% (1:76). There was no gender predominance. Ectopic uretrocele was seen in 6 (1.5%) fetuses, and dysplastic kidney of one of the poles was diagnosed in 5 (1.3%) fetuses, 3 of which had an ectopic ureterocele (Chart Flow). Eleven fetuses (2.8%) had associated anomalies, and the pregnancy was terminated per parental request. Four of those had chromosomal abnormalities; two fetuses had Trisomy 18, one with Triploid, and one with Williams syndrome.

One hundred and fourteen women returned to our clinic in their subsequent pregnancy and follow up on their infants with isolated DCS (57 boys and 57 girls) was provided. The follow up was performed between 1.5 and 8 years after the initial diagnosis of DCS in the previous pregnancy, with a median of 4 years. Of these, 95 cases had a postnatal renal ultrasound performed, and the diagnosis of DCS was confirmed in all of them (chart flow). One girl (1/95, 1%) was diagnosed with urinary reflux with recurrent UTIs, and two boys (2/57, 3.5%) were diagnosed with undescended testis. In 10/114 (8.7%) women, DCS was diagnosed in their subsequent pregnancy as well.

## 4. Discussion

The renal double-collecting system in the fetus has been addressed in previous studies [3,7,8], but there is little data regarding its association with genetic syndromes. One study examining the association between isolated prenatally diagnosed DCS and genetic aberrations in chromosomal microarray analyses (CMA) [6] found a rate of 1.4% (2/143) of pathogenic CMA, which is not significantly different from the risk in the general population. In our study, a genetic abnormality was found among 3% of fetuses with a renal DCS who also had other major malformations leading to the termination of pregnancy. That amount is also similar in the general population. No chromosomal abnormalities were reported in 114 newborns diagnosed with an isolated renal DCS for whom postnatal follow up was obtained. Our data support the finding by Singer et al. [6], concluding that the prenatal diagnosis of isolated renal DCS is not associated with an increased rate of fetal chromosomal aberrations.

Our data, in concordance with other studies [6,10], concluded that the prenatal diagnosis of isolated renal DCS is not associated with an increased rate of fetal chromosomal aberrations.

We found a rate of 1:76 (1.3%) of DCS in low-risk pregnancies, which is slightly higher compared to previous studies reporting a rate of 0.8–1% [8,11,12]. This may be attributed both to our transvaginal approach as well as to the timing of the exam in early pregnancy as opposed to a transabdominal approach later in pregnancy used in other studies. The increased renal echogenicity typical of early pregnancy helps the sonographer visualize the normal and abnormal kidney shape, and the high-resolution transvaginal probe enhances the visualization of the eight-digit kidney shape and double ureters (Figure 1, Figure 2 and Figure 3). In contrast to other studies reporting female predominance [13,14], we did not find sex differences consistent with Avni et al. [15]. However, unlike studies reporting the incidence of DCS during infancy or childhood, diagnosed mostly in symptomatic (e.g., UTI) and some incidental cases, prenatal screening might reveal cases that would otherwise not be diagnosed. Further epidemiological studies are required to clarify this issue.

An interesting finding in our cohort was the recurrence of DCS in 8% of subsequent pregnancies. A previous study [16] exploring the familial clustering of different congenital anomalies of the kidney and urinary tract among families in Turkey found a much higher rate of occurrence [18% (2/11)] in the siblings of patients with DCS. However, the series was small (*n* = 11), and the index patients with DCS in this study were identified in a hospital setting, while our population of fetuses represented a form of “screening” of the general population, irrespective of the presence or absence of symptoms.

Ectopic uretrocele was diagnosed in 1.5% of fetuses in our cohort compared to other studies reporting higher rates of associated renal anomalies [4,7,8,17,18]. This difference could be explained by variations in gestational age and the sonographic examination approach. In transabdominal US, the physician looked for accompanied prominent markers like ectopic urotrocele or hydroureter to enhance diagnosis accuracy.

One (1%) female child in our cohort was diagnosed with vesicoureteral reflux and recurrent UTIs, a rate similar to that of the general population and in agreement with other studies that reported a low rate of urinary vesicoureteral reflux complicated by UTIs among children [18,19,20,21,22]. Interestingly, a meta-analysis, which included 11 studies with 284 fetuses who were diagnosed with isolated DCS [23], reported a significant proportion of vesicoureteral reflux and urinary tract infections postnatally. However, renal pelvis dilatation was present in many cases, and 41% of the fetuses had worsening pelvic/ureteric dilatation over the course of the pregnancy, which could affect the outcome. In addition, DCS diagnosis in our cohort was made exclusively in early pregnancy as opposed to diagnosis late in pregnancy when severe changes in renal structure were more likely to be detected compared to more subtle anomalies. Although we did not address this issue of the fetal renal sonographic abnormalities that can appear later in pregnancy, such as hydronephrosis, we concur with Hopkins et al. [10] that duplicated collecting systems should be monitored with an ultrasound examination in the third trimester to evaluate for potential hydronephrosis in one or both renal pelvises.

DCS and cryptorchidism have not been linked yet in the literature. Cryptorchidism has a diversity of causes [24,25], with a prevalence varying between 1 and 9% for at least one cryptorchidic testis [25,26]. Our finding of 3.5% (2/57 boys) undescended testis in boys with DSC indicates that renal DCS does not increase the risk of cryptorchidism.

The strengths of our study include the performance of the exams using the transvaginal method in early pregnancy by a single expert sonographer, thus maximizing accuracy, as well as our relatively large cohort. Our limitations include our selective follow up of less than a third of the cohort, as we did not obtain the postnatal follow up of women who did not return to our clinic, and possible recall bias was inherent in retrospective reports due to the systematic difference in the ability of participant groups to accurately recall information. In addition, the newborns were not examined after birth. It should be mentioned that neonates with renal DCS are often free of clinical symptoms and most probably do not undergo any test to verify the diagnosis. To approve the false-negative rate postnatally, the screening investigation could include radioisotope tests; this approach is neither justified nor practical.

## 5. Conclusions

In conclusion, no genetic aberrations were found in fetuses diagnosed with the isolated renal double-collecting system, and the likelihood of other anomalies and chromosomal aberrations was similar to the general population. No increase in the rate of vesicoureteral reflux and cryptorchidism were found. There was a recurrence rate of 8% in subsequent pregnancies. In light of its benign nature, we speculate that renal DCS may be considered an anatomic variant rather than a major abnormal fetal sonographic finding, necessitating further investigations. A repeat ultrasound examination in the third trimester to detect renal abnormalities that develop late in pregnancy, including hydronephrosis, should be incorporated into the follow up protocol for fetuses diagnosed with renal DCS.

We are of the opinion that these results should be presented during genetic counseling to patients referred for counseling due to a double-collecting system. Prospective, well-adjusted studies are still needed to determine the optimal management of pregnancies with this malformation.

## Figures and Tables

**Figure 1 jcm-12-07124-f001:**
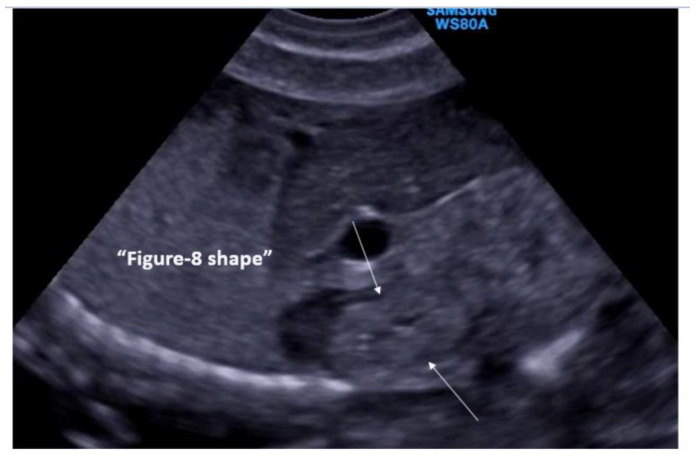
Visualization of the 8-digit kidney shape indicating the presence of the renal double-collecting system.

**Figure 2 jcm-12-07124-f002:**
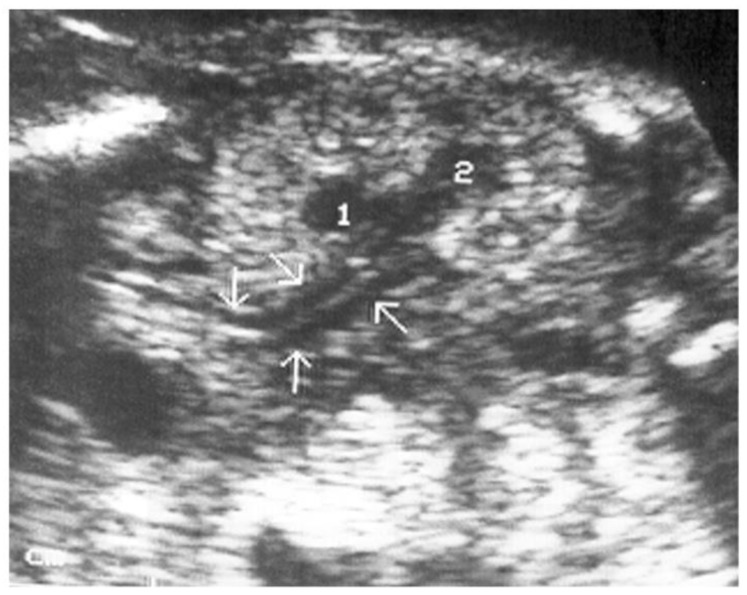
The eight-kidney shape sign (two renal pelvis 1 and 2) with enlarged ureters (arrows) at 15 gestational age indicating the presence of the renal double-collecting system.

**Figure 3 jcm-12-07124-f003:**
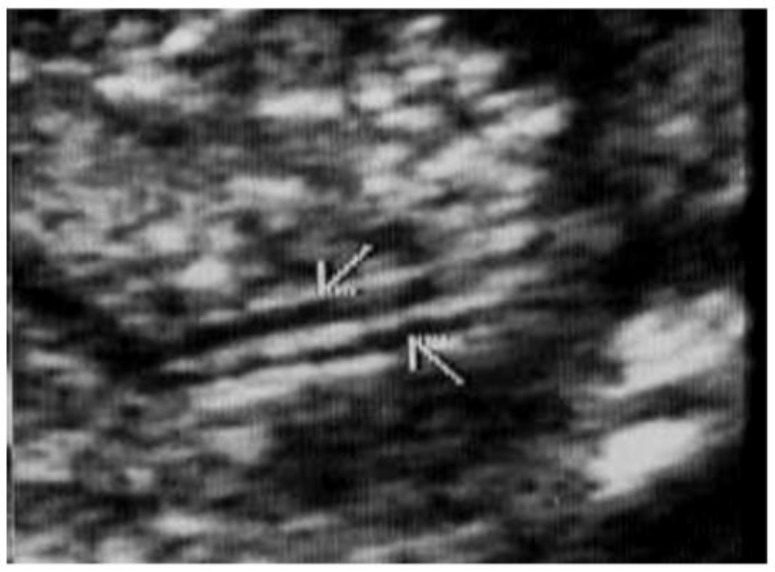
Visualization of two ureters (arrows).

## Data Availability

Data are unavailable due to privacy.

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
