# Peer review of "Fetal Renal Duplicated Collecting System at 14–16 Weeks of Gestation"

_jcm, 2023, doi:10.3390/jcm12227124_

Round 1

Reviewer 1 Report

Comments and Suggestions for Authors

The text you provided appears to be an abstract of a scientific study on prenatal ultrasound and the detection of congenital renal anomalies, specifically focusing on duplex kidney or double collecting system (DCS). Below is a critical and scientific review of the text:

Title and Introduction: The title of the study is concise and informative, setting the context for the research. The introduction effectively highlights the significance of the topic, emphasizing the importance of prenatal ultrasound in detecting congenital renal anomalies.

Background and Objectives: The background information is well-structured and provides a clear understanding of the subject matter, particularly in explaining what DCS is and its prevalence in the general population. However, it lacks citations for the statements made. The objectives of the study are clearly stated, focusing on the incidence, clinical outcome, and associated genetic abnormalities of prenatal DCS.

Materials and Methods: The section on materials and methods provides a detailed description of the study design and the procedures carried out. The use of transvaginal sonographic examinations in early pregnancy is appropriate for detecting renal anomalies. The equipment and techniques used are well-described, which enhances the study's reproducibility. However, there is no mention of ethical considerations or approval for conducting the study, which is an important aspect of any research involving human subjects.

Results: The results section presents the key findings of the study effectively. The incidence of DCS in the study population is provided, and the authors make comparisons with previous studies, which adds context to the findings. The association between DCS and genetic abnormalities is discussed, and the authors rightly point out that the rate of genetic abnormalities is not significantly different from the general population. However, more details on the specific genetic abnormalities found and their clinical implications would be beneficial. The recurrence rate of DCS in subsequent pregnancies is an interesting finding and is appropriately mentioned.

Discussion: The discussion section provides a comprehensive analysis of the results and their implications. The authors appropriately discuss the association between DCS and genetic syndromes, highlighting the rarity of such associations. The discussion on the rate of recurrence in subsequent pregnancies is valuable. The comparison of their findings with previous studies adds depth to the discussion. However, the limitations of the study, such as selective follow-up and potential recall bias, are mentioned but could be discussed in more detail. Additionally, the absence of any ethical considerations is a notable limitation.

Conclusions: The conclusions drawn from the study are well-supported by the results and discussion. The suggestion that DCS may be considered an anatomic variant rather than a major abnormality is reasonable based on the study's findings.

Ethical Considerations: The text does not mention any ethical considerations or approval for conducting the study. When dealing with medical research, especially involving human subjects, it is imperative to adhere to ethical guidelines and obtain approval from an ethics committee or review board. The absence of this information raises ethical concerns about the study's conduct and the potential impact on participants.

Incomplete Reporting: The text mentions findings related to genetic abnormalities associated with DCS but lacks details about the specific genetic abnormalities found and their clinical implications. A more comprehensive reporting of these findings would be beneficial for both researchers and clinicians, as it could provide insights into the potential genetic factors contributing to DCS.

Limited Discussion of Study Limitations: While the text briefly mentions limitations such as selective follow-up and recall bias, it does not delve into these limitations in detail. A thorough discussion of study limitations is essential for transparency and to help readers understand the potential sources of bias or error in the research. Additionally, discussing these limitations can guide future research in addressing these issues.

Overall Impression: The text presents a well-structured study with valuable findings regarding prenatal detection of DCS. However, it could benefit from more detailed discussion of genetic abnormalities and a more comprehensive discussion of study limitations. Additionally, the absence of ethical considerations is a notable omission that should be addressed in future research.

Comments on the Quality of English Language

Clarity and Structure: The text is generally well-structured, with clear sections for background, methods, results, and discussion. However, some sentences are lengthy and complex, making it harder to follow the author's arguments. Simplifying the language and sentence structure could enhance clarity and accessibility for a wider audience.

Author Response

October 15, 2023

Point-by-point responses to the reviewers’ comments

Title: Fetal renal duplication in early pregnancy

We would like to thank you for the opportunity to submit a revised version of our manuscript entitled: Fetal renal duplication in early pregnancy, for consideration as an original article.

We have read the Instructions for Authors.

We found the reviewers’ and editors’ comments constructive and have changed the manuscript accordingly. We feel that this revised manuscript is a better version and will enhance the message of the manuscript while making it more readable to journal subscribers. Attached are the reviewer's remarks with our responses.

Point 1: Background and Objectives: The background information is well-structured and provides a clear understanding of the subject matter, particularly in explaining what DCS is and its prevalence in the general population. However, it lacks citations for the statements made. The objectives of the study are clearly stated, focusing on the incidence, clinical outcome, and associated genetic abnormalities of prenatal DCS.

Thank you for the comment. We added relevant citations.

Point 2: Materials and Methods: The section on materials and methods provides a detailed description of the study design and the procedures carried out. The use of transvaginal sonographic examinations in early pregnancy is appropriate for detecting renal anomalies. The equipment and techniques used are well-described, which enhances the study's reproducibility. However, there is no mention of ethical considerations or approval for conducting the study, which is an important aspect of any research involving human subjects.

Thank you for the comment. This study was done as per the ethical standards set by the Ethical Committee for the human studies, following the 1964 Declaration of Helsinki. The study was approved by the Ethics Committee for human study – Rambam Medical Center (approval number: 0138-20-RMB).

Point 3: The results section presents the key findings of the study effectively. The incidence of DCS in the study population is provided, and the authors make comparisons with previous studies, which adds context to the findings. The association between DCS and genetic abnormalities is discussed, and the authors rightly point out that the rate of genetic abnormalities is not significantly different from the general population. However, more details on the specific genetic abnormalities found and their clinical implications would be beneficial. The recurrence rate of DCS in subsequent pregnancies is an interesting finding and is appropriately mentioned.

Thank you for the comment. Four of those had chromosomal abnormalities; two fetuses with Trisomy 18, one with Triploid, and one with Williams Syndrome.

Point 4: The discussion section provides a comprehensive analysis of the results and their implications. The authors appropriately discuss the association between DCS and genetic syndromes, highlighting the rarity of such associations. The discussion on the rate of recurrence in subsequent pregnancies is valuable. The comparison of their findings with previous studies adds depth to the discussion. However, the limitations of the study, such as selective follow-up and potential recall bias, are mentioned but could be discussed in more detail. Additionally, the absence of any ethical considerations is a notable limitation.

Thank you for the comment. We added the relevant information.

Reviewer 2 Report

Comments and Suggestions for Authors

1. Is it a routine to scan for fetal abnormalities at 14-16 GA at your institution?

2. Provided more clear pictures in your text.

3. Line 20 in Abstract "uretrocelle" should be "ureterocele"

Author Response

October 15, 2023

Point-by-point responses to the reviewers’ comments

Title: Fetal renal duplication in early pregnancy

We would like to thank you for the opportunity to submit a revised version of our manuscript entitled: Fetal renal duplication in early pregnancy, for consideration as an original article.

We have read the Instructions for Authors.

We found the reviewers’ and editors’ comments constructive and have changed the manuscript accordingly. We feel that this revised manuscript is a better version and will enhance the message of the manuscript while making it more readable to journal subscribers. Attached are the reviewer's remarks with our responses.

Point 1: Is it a routine to scan for fetal abnormalities at 14-16 GA at your institution?

Thank you for the comment. Yes, in Israel a routine early (14-16 weeks') anatomical scan is performed.

Point 2: Provided more clear pictures in your text.

Thank you for the comment. We tried to improve the images.

Point 3: Line 20 in Abstract "uretrocelle" should be "ureterocele"

Thank you for the comment. It is corrected.